# In Patients with Severe COVID-19, the Profound Decrease in the Peripheral Blood T-Cell Subsets Is Correlated with an Increase of QuantiFERON-TB Gold Plus Indeterminate Rates and Reflecting a Reduced Interferon-Gamma Production

**DOI:** 10.3390/life12020244

**Published:** 2022-02-07

**Authors:** Alessandra Imeneo, Grazia Alessio, Andrea Di Lorenzo, Laura Campogiani, Alessandra Lodi, Filippo Barreca, Marta Zordan, Virginia Barchi, Barbara Massa, Simona Tedde, Angela Crea, Pietro Vitale, Ilaria Spalliera, Mirko Compagno, Luigi Coppola, Luca Dori, Vincenzo Malagnino, Elisabetta Teti, Massimo Andreoni, Loredana Sarmati, Marco Iannetta

**Affiliations:** 1Department of System Medicine, University of Rome Tor Vergata, Via Montpellier 1, 00133 Rome, Italy; alessandra.imeneo@students.uniroma2.eu (A.I.); grazia.alessio@students.uniroma2.eu (G.A.); andrea.dilorenzo@students.uniroma2.eu (A.D.L.); alessandra.lodi@students.uniroma2.eu (A.L.); filippo.barreca@alumni.uniroma2.eu (F.B.); marta.zordan@students.uniroma2.eu (M.Z.); virginia.barchi@students.uniroma2.eu (V.B.); barbara.massa@alumni.uniroma2.eu (B.M.); simona.tedde@students.uniroma2.eu (S.T.); angela.crea@ptvonline.it (A.C.); vincenzo.malagnino@ptvonline.it (V.M.); andreoni@uniroma2.it (M.A.); srmldn00@uniroma2.it (L.S.); 2Infectious Disease Clinic, Policlinico Tor Vergata, Viale Oxford 81, 00133 Rome, Italy; laura.campogiani@ptvonline.it (L.C.); pietro.vitale@ptvonline.it (P.V.); ilaria.spalliera@ptvonline.it (I.S.); mirko.compagno@ptvonline.it (M.C.); luigi.coppola@ptvonline.it (L.C.); luca.dori@ptvonline.it (L.D.); elisabetta.teti@ptvonline.it (E.T.)

**Keywords:** QuantiFERON-TB Gold, IGRA, SARS-CoV-2, CD3, lymphopenia, IFN

## Abstract

Increased rates of indeterminate QuantiFERON-TB Gold Plus Assay (QFT-Plus) were demonstrated in patients hospitalized with Coronavirus Disease (COVID)-19. We aimed to define the prevalence and characteristics of hospitalized COVID-19 patients with indeterminate QFT-Plus. A retrospective study was performed including hospitalized COVID-19 patients, stratified in survivors and non-survivors, non-severe and severe according to the maximal oxygen supply required. Statistical analysis was performed using JASP ver0.14.1 and GraphPad Prism ver8.2.1. A total of 420 patients were included, median age: 65 years, males: 66.4%. The QFT-Plus was indeterminate in 22.1% of patients. Increased rate of indeterminate QFT-Plus was found in non-survivors (*p* = 0.013) and in severe COVID-19 patients (*p* < 0.001). Considering the Mitogen-Nil condition of the QFT-Plus, an impaired production of interferon-gamma (IFN-γ) was found in non-survivors (*p* < 0.001) and in severe COVID-19 patients (*p* < 0.001). A positive correlation between IFN-γ levels in the Mitogen-Nil condition and the absolute counts of CD3+ (*p* < 0.001), CD4+ (*p* < 0.001), and CD8+ (*p* < 0.001) T-lymphocytes was found. At the multivariable analysis, CD3+ T-cell absolute counts and CD4/CD8 ratio were confirmed as independent predictors of indeterminate results at the QFT-Plus. Our study confirmed the increased rate of indeterminate QFT-Plus in COVID-19 patients, mainly depending on the peripheral blood T-lymphocyte depletion found in the most severe cases.

## 1. Introduction

In December 2019, the first cases of pneumonia caused by a new coronavirus, named Severe Acute Respiratory Syndrome Coronavirus-2 (SARS-CoV-2), was reported in Wuhan, province of Hubei, China, and quickly spread to the rest of the world, causing the Coronavirus Disease (COVID)-19 pandemic. [1,2,3]. The virus can cause a wide variety of clinical manifestations, from asymptomatic disease to severe pneumonia with respiratory failure, requiring mechanical ventilation. Several risk factors have been associated with severe forms of SARS-CoV-2 infection. Among these, older age, male sex, and comorbidities seem to play a major role [3]. Nonetheless, despite several studies on pathophysiological processes, viral and host factors responsible for patient morbidity and mortality remain partially unknown [4].

SARS-CoV-2 pathogenesis is associated with the lung damage and the procoagulant state triggered by an exaggerated immune response characterized by the so-called “cytokine storm”; furthermore, an increased mortality has been associated to a dysregulated host immune response [4,5,6,7,8,9,10,11].

Considering the association between “cytokine storm” and severity of COVID-19, immunomodulating drugs, such as corticosteroids and anti-interleukin (IL)-6 receptor monoclonal antibodies, have been approved for the treatment of this disease [12,13,14,15]. As these are immunosuppressive agents, before or during treatment patients are usually screened for latent infections such as Mycobacterium tuberculosis (MTB) [16,17,18]. 

The QuantiFERON-TB Gold Plus (QFT-Plus) assay (Qiagen) is largely utilized to identify latent tuberculosis, and is based on the detection of interferon-gamma (IFN-γ) released by T-lymphocytes, after in vitro stimulation of human whole-blood with specific antigens of the MTB complex. The test consists of four vacutainer tubes: two contain MTB peptides for the specific stimulation of CD4+ (TB1 tube) and CD4+ and CD8+ T-lymphocytes (TB2 tube); the third (Mitogen) uses phytohemagglutinin (PHA) as a positive control to assess the overall efficacy of host immune response, and the last one (Nil) is the negative control, to evaluate the basal IFN-γ production, without stimulation. The PHA positive control (Mitogen) is essential to identify those patients for whom the result of the test cannot be considered reliable, due to a global impairment in the activation capacity of T-lymphocytes, even with a non-specific stimulation (indeterminate result) [19,20]. Therefore, the Mitogen (and Mitogen-Nil) condition in the QFT-Plus assay is a measure of the nonspecific T-cell stimulation and an indicator of the immune dysregulation.

Several studies have shown an increased number of indeterminate QFT assay results in patients hospitalized because of SARS-CoV-2 [16,17,21,22,23,24,25]. Patients with an indeterminate response are those with compromised immune responses and inadequate ability to be activated by a mitogen control [21,24]. Furthermore, indeterminate results of the QFT assay could predict mortality in COVID-19 patients [21]. In a recent work on a limited number of subjects, it has been shown that patients with severe COVID-19 had a decreased IFN-γ production after mitogen stimulation compared to asymptomatic COVID-19 patients and healthy donors, in an endemic area for tuberculosis [25].

Both altered immune response and lymphopenia might account for QTF-Plus assay indeterminate results. In COVID-19, lymphopenia has been associated with severe forms of SARS-CoV-2 [4,26] and the absolute peripheral T-cell count has been recognized as a possible tool to identify patients at risk of poor outcomes [27]. 

Our study aimed to define the prevalence of indeterminate QFT-Plus assay in patients hospitalized for SARS-CoV-2 infection, to evaluate the association with the outcome and severity of the disease, and to identify the factors influencing the indeterminate results of the QFT-Plus assay.

## 2. Materials and Methods

### 2.1. Study Design, Eligibility Criteria, and Data Collection

This was a retrospective, observational study performed at the University Hospital Policlinico Tor Vergata of Rome, Italy, including adult patients (>18 years) hospitalized because of SARS-CoV-2 infection in the Infectious Disease Clinic between March and May 2020 (first wave) as well as September and December 2020 (second wave). 

The study was approved by the Ethics Committee of Fondazione PTV Policlinico Tor Vergata (Protocol number 154/21). All procedures were carried out in accordance with the Declaration of Helsinki (WMA 2013) and relevant guidelines and regulations. Considering the retrospective nature of the study the requirement for informed consent was waived by the Ethics Committee, as permitted by the local legislation.

In our hospital, QTF-Plus assay was routinely performed in all COVID-19 hospitalized patients to identify the presence of latent tuberculosis infection (LTBI). On hospital admission, peripheral blood lymphocyte absolute counts were also routinely assessed in COVID-19 patients. Both tests were used to better characterize COVID-19 patients, considering the possibility of starting a treatment with anti-IL-6 receptor and other immunomodulatory therapies.

Patients who fulfilled the following criteria were enrolled in the study: 1. hospitalized at the infectious diseases ward due to SARS-CoV-2 infection, as documented by a positive reverse transcription-polymerase chain reaction (RT-PCR) for SARS-CoV-2 RNA detection on a nasopharyngeal swab (NPhS), either symptomatic or asymptomatic; 2. QFT-Plus assay performed during hospitalization within 30 days from the first positive NPhS. Asymptomatic patients were hospitalized in the infectious disease ward because of concomitant clinical conditions that required inpatient care associated with SARS-CoV-2 transmission prevention, although they did not show any sign or symptom related to SARS-CoV-2 infection. In the manuscript, both symptomatic and asymptomatic patients are referred to as COVID-19.

During the study period, demographics, clinical, and laboratory data of hospitalized COVID-19 patients were collected. The following information were recorded in an electronic ad hoc created database: age, sex, outcome, need for Intensive Care Unit (ICU), comorbidities, date of symptom onset, and first positive RT-PCR for SARS-CoV-2 on a NPhS, oxygen support, treatments, QFT-Plus assay results, peripheral blood lymphocyte subset assessment, neutrophil and lymphocyte counts, inflammation-related parameters (C-reactive protein (CRP), IL-6, tumor necrosis factor (TNF)-α, fibrinogen, ferritin, D-dimer). Neutrophils and lymphocyte counts, as well as inflammation-related parameters registered in the database and included in the statistical analysis were performed on admission to the Infectious Disease ward. 

All blood tests were performed in the hospital central laboratory following standard procedures. T- (CD3+), B- (CD19+), and Natural Killer (NK)- (CD3negCD16+CD56+) lymphocyte subsets were assessed by multiparametric flow cytometry on peripheral blood samples, according to standardized procedures, as previously described [27].

### 2.2. Definitions

The QFT-Plus assay consists of four vacutainer tubes: TB1 detects CD4+ T-cell response; TB2 is optimized for detection of CD4+ and CD8+ T-cell responses; the Mitogen tube represents the positive control, while the Nil tube represents the negative control. 

In a typical QFT-Plus assay, the level of INF-γ in the Nil condition should be <8.0 IU/mL, and is subtracted to the other conditions (TB1-Nil, TB2-Nil, Mitogen-Nil). The test gives 3 possible results: a. positive when INF-γ level in Nil condition is ≤8.0 IU/mL and in TB1-Nil and/or TB2-Nil is ≥0.35 IU/mL and ≥25% of Nil; b. negative when INF-γ level in Nil condition is ≤8.0 and in Mitogen-Nil is ≥0.5 and in both TB1-Nil and TB2-Nil INF-γ levels are either ≤0.35 IU/mL or ≥0.35 IU/mL but ≤25% of Nil; and c. indeterminate when Nil is >8.0 IU/mL or Nil is ≤8.0 IU/mL and Mitogen-Nil < 0.5 IU/mL and in both TB1-Nil and TB2-Nil INF-γ levels are either ≤0.35 IU/mL or ≥0.35 IU/mL but ≤25% of Nil [20]. Positive and negative results to the QFT-Plus were considered as determinate results, in contrast to indeterminate results.

According to the outcome, patients were divided into non-survivors, if death occurred during hospitalization, or survivors, if they were discharged.

As for oxygen support, five groups were identified, based on maximal oxygen support required during the hospitalization; ambient air (AA), Venturi Oxygen Mask (VMK), Non-Rebreather Mask (NRM), Non-Invasive Ventilation (NIV), or Orotracheal Intubation (OTI) for mechanical ventilation. Oxygen groups were linked to disease severity, and patients were classified as non-severe (AA and VMK) and severe (NRM, NIV, and OTI).

COVID-19 patients on oxygen-therapy received steroid treatment according to national and international guidelines available at the time of hospitalization. Furthermore, a number of patients were treated with anti-IL-6 receptor monoclonal antibodies, following the Italian Medicines Agency (AIFA) guidelines and the evidence from international literature [28,29,30].

### 2.3. Statistical Analysis

All data were analyzed using JASP (Version 0.14.1) and GraphPad Prism (Version ver8.2.1). Categorical variables are presented as absolute frequency and percentages (%), while quantitative variables are presented as medians and interquartile ranges (IQR). Differences between groups were assessed with the two-tailed Chi-square test for categorical data, and the non-parametric Mann–Whitney test (2 groups) or Kruskal–Wallis test (>2 groups) for quantitative data. Correlations were assessed using the non-parametric Spearman’s test. Multivariable regression analysis was used to assess the factors associated to QFT-Plus indeterminate results. With this aim, all the factors significantly associated with an indeterminate QFT-Plus assay at the univariable analysis, were included in the multivariable regression model. 

For all the tests, the level of statistical significance was <0.05. 

## 3. Results

### 3.1. Study Population

During the study period, 424 patients hospitalized for COVID-19 in the Infectious Disease Clinic of the University Hospital Policlinico Tor Vergata of Rome, were included in the study. A total of 150 were hospitalized from March to May 2020 (First Wave of COVID-19) and 274 were hospitalized from September to December 2020 (Second Wave of COVID-19). All patients had at least one positive RT-PCR for SARS-CoV-2 RNA detection on a NPhS. Four patients were excluded from the analysis because the QFT-Plus assay was performed more than 30 days after the first positive SARS-CoV-2 NPhS; therefore, 420 patients were evaluated. For 18 patients, peripheral blood T-, B-, NK-lymphocyte absolute counts were not available.

Characteristics of the study population are reported in Table 1. Overall, median age was 65 years (IQR: 52–77), with a prevalence of male sex (279; 66.4%) and symptomatic patients (400; 95.2%). Charlson comorbidity index median value was 4 (IQR: 2–5). The rate of ICU admission was 19.8% (83/420), with a mortality rate of 23.1% (97/420). 

After stratifying patients in survivors vs. non-survivors and non-severe vs. severe, male sex, older age, higher Charlson index were associated with a worse outcome (*p* = 0.010, *p* < 0.001, and *p* < 0.001, respectively) and a more severe disease (*p* < 0.001, *p* < 0.001, and *p* < 0.001, respectively). Notably, cardiovascular, pulmonary, neurological/psychiatric, renal, cerebrovascular, hematological disorders, diabetes, and solid tumors were associated with an increased mortality (*p* < 0.001, *p* < 0.001, *p* < 0.001, *p* < 0.001, *p* < 0.001, *p* < 0.001, *p =* 0.017, and *p* = 0.021, respectively). Cardiovascular, pulmonary, hematological, and immunological/rheumatological disorders were also associated with a more severe disease (*p* = 0.005, *p* = 0.038, *p* = 0.004, and *p* = 0.010, respectively). ICU admission rate was higher in non-survivors and severe patients (*p* < 0.001 and *p* < 0.001, respectively) (Table 1).

The median number of days between the onset of symptoms and the first positive NPhS for SARS-CoV-2 RNA detection was five days (IQR 1–8), while the median number of days between the onset of symptoms and the blood sampling for the QFT-Plus assay was 10 days (IQR 7–14). Median days from the first positive NPhS for SARS-CoV-2 RNA detection and the blood sampling for the QFT-Plus assay was four days (IQR 2–7). For 20 patients, the exact onset of symptoms was unknown, or they were asymptomatic for SARS-CoV-2 related symptoms. In these cases, hospitalization in the infectious disease ward was due to the presence of concomitant diseases (pneumological, hematological, solid tumors, myocardial infarction, cerebrovascular accidents) requiring inpatient care, or the need of hemodialysis, in a setting where SARS-CoV-2 containment protocols have been adopted. Statistical analyses after excluding asymptomatic patients are shown in Appendix A. Peripheral blood T-, B- and NK-lymphocyte assessment was performed on the same day of the QFT-Plus assay (median elapsed time [IQR] between the two tests: 0.0 (0.0–3.0) days) (Table 1).

### 3.2. Laboratory Parameters and Inflammation Markers

The analysis of inflammation markers assessed at baseline (on admission to the Infectious Disease Ward) showed increased levels of CRP (60.1 mg/L, IQR: 25.1–113.5 mg/L), D-dimers (942 ng/mL, IQR: 519–1601 ng/mL), fibrinogen (550 mg/dL, IQR: 418–684 mg/dL), and ferritin (635 ng/mL, IQR: 282–1252 ng/mL) in the overall cohort of COVID-19 patients, compared to the reference values of the central laboratory (Table 2). 

After stratifying patients according to the outcome and disease severity, lymphocyte counts were reduced in non-survivors and severe patients compared to survivors and non-severe patients (*p* < 0.001 and *p* < 0.001, respectively). Conversely neutrophils counts were increased in non-survivors and severe patients compared to survivors and non-severe patients (*p* < 0.001 and *p* < 0.001, respectively). Therefore, the neutrophil-to-lymphocyte (N/L) ratio was significantly increased in non-survivors and severe patients compared to survivors and non-severe patients (*p* < 0.001 and *p* < 0.001, respectively). CRP, IL-6, D-dimers, fibrinogen, and ferritin were significantly different after comparing survivors with non-survivors and non-severe with severe COVID-19 patients (Table 2). Although TNF-α did not significantly differ after comparing survivors vs. non-survivors, its levels were lower in severe than non-severe COVID-19 patients (*p* < 0.001) (Table 2).

### 3.3. QFT-Plus Assay Results

The QFT-Plus assay results were defined as determinate (either positive or negative) in 327 patients, with 292 (70%) negative and 35 (8%) positive results. A total of 93 patients (22%) had an indeterminate QFT-Plus assay. After stratifying patients according to the outcome and disease severity, higher rates of indeterminate results were found in non-survivors compared to survivors COVID-19 patients (32.0% vs. 19.2%, *p* = 0.008, respectively) and in severe compared to non-severe COVID-19 patients (33.8% vs. 10.8% *p* < 0.001, respectively).

Median values of IFN-γ production for TB1 and TB1-Nil, TB2 and TB2-Nil, Mitogen and Mitogen-Nil tubes of the QFT-Plus assay are reported in Table 3. Overall, reduced levels of IFN-γ production were found in TB1, TB1-Nil, TB2 and TB2-Nil conditions in non-survivors and severe patients compared to survivors and non-severe COVID-19 patients (Table 3).

We focused on the Mitogen-Nil condition, and significantly lower levels of IFN-γ production were found in non-survivors compared to survivors (1.3 IU/mL vs. 5.5 IU/mL, *p* < 0.001) and in severe compared to non-severe patients (1.7 IU/mL vs. 8.4 IU/mL, *p* < 0.001) (Table 3).

After stratifying patients according to the maximal oxygen supply received during hospitalization, the rates of indeterminate QFT-Plus assay were significantly increased in patients treated with non-invasive or invasive ventilation compared to patients treated with Venturi Mask or not needing oxygen supplementation (QFT-Plus determinate: 38.5%, 19.6%, 7.3%, 23.2%, and 11.3%; QFT-Plus indeterminate: 4.3%, 20.4%, 11.8%, 37.6%, and 25.8%, for AA, VMK, NRM, NIV, and OTI, respectively; Chi^2^ test *p* < 0.001) (Figure 1). Furthermore, a decreasing trend of IFN-γ production in the Mitogen-Nil condition was found proceeding from patients who did not need oxygen therapy (AA, 9.63 IU/mL), through patients with VMK (2.68 IU/mL), NRM (1.90 IU/mL) and NIV (1.92 IU/mL), to patients who underwent OTI (0.97 IU/mL) for mechanical ventilation (*p* < 0.001). Additionally, similar results were obtained after removing from the analysis patients with an indeterminate result at the QFT-Plus assay (*p* < 0.001). Thus, increasing disease severity was associated with decreasing IFN-γ levels produced in the Mitogen-Nil condition of the QFT-Plus assay (Figure 1). 

### 3.4. QFT-Plus Mitogen-Nil Results and Correlation with Neutrophil and Lymphocyte Counts, Lymphocyte Subset Absolute Counts and Inflammation Markers

COVID-19 patients were stratified according to the QFT-Plus assay results in determinate and indeterminate. Table 4 reports characteristics of the two subpopulations. Patients with an indeterminate QFT-Plus were prevalently males (*p* = 0.041), older (*p* = 0.003), and showed higher ICU admission rates (*p* = 0.002) than patients with a determinate result. Importantly, time of blood sampling for QFT-Plus, T-, B-, NK-lymphocyte subset assessment, and NPhS for SARS-CoV-2 RNA detection did not significantly differ between the two groups (Table 4).

Considering laboratory parameters, neutrophils count (*p* < 0.001), N/L ratio (*p* < 0.001), CRP (*p* < 0.001), D-dimers (*p* < 0.001), fibrinogen (*p* < 0.001), ferritin (*p* < 0.001), and CD4/CD8 ratio (*p* < 0.001) were significantly increased in patients with an indeterminate result compared to patients with a determinate result at the QFT-Plus assay. Conversely, total lymphocyte counts (*p* < 0.001), total CD3+ (*p* < 0.001) and CD3+CD4+ (*p* < 0.001), CD3 + CD8+ (*p* < 0.001) T-lymphocyte, as well as CD3negCD16 + CD56+ NK-lymphocyte (*p* < 0.001) absolute counts were reduced in patients with an indeterminate result compared to patients with a determinate result at the QFT-Plus assay. No differences in IL-6 levels were found after comparing the two groups, despite the increased levels of IL-6 previously detected in non-survivors and severe COVID-19. A slight decrease in TNF-α levels were found in those patients with an indeterminate compared to a determinate QFT-Plus assay (10.2 pg/mL vs. 14.8 pg/mL, *p* < 0.001) (Table 4). 

The levels of IFN-γ in the Mitogen-Nil condition were positively correlated with total lymphocyte absolute counts, total CD3+, CD3+CD4+, CD3+CD8+, CD3negCD16+CD56+ absolute counts, and TNF-α plasmatic levels (Spearman’s test *p* < 0.0001 for all the correlations). An inverse correlation between IFN-γ levels in the Mitogen-Nil condition and CD4/CD8 ratio, neutrophil counts, N/L ratio, CRP, D-dimers, fibrinogen and ferritin was also found (Spearman’s test *p* < 0.0001 for all the correlations) (Figure 2). 

### 3.5. Steroid-Treatment and Anti-IL6 Receptor Monoclonal Antibodies

Overall, during hospitalization, 262 out of 420 patients (62.4%) received steroid-treatment and 41 out of 420 patients (9.8%) received anti-IL-6 receptor monoclonal antibodies (32 tocilizumab and 9 sarilumab). Thirty-six subjects received treatment before blood sampling for QFT-Plus, specifically 18, 13, and 3 patients received steroid, anti-IL-6 receptor monoclonal antibodies, and both the treatment, respectively. For two patients, the timing of immune modulating treatment was unavailable. After removing these subjects (36 individuals) from the analysis (384 subjects left), the rates of indeterminate results were confirmed to be significantly increased in non-survivors compared to survivors (29/90 (32.2%) vs. 54/294 (18.4%), *p* = 0.005) and in severe compared to non-severe COVID-19 patients (64/183 (35.0%) vs. 19/201 (9.5%), *p* < 0.001), corroborating previous results observed in the overall cohort. 

### 3.6. Multivariable Analysis

After demonstrating that patients with severe disease and poor outcome had an increased rate of indeterminate response at the QFT-Plus assay, we aimed to identify which were the factors associated to this phenomenon. At the univariable analysis, male sex (*p* = 0.042), age > 65 (*p* = 0.008), a higher Charlson Index (*p* = 0.022), increased values of N/L ratio (*p* < 0.001), CRP (*p* < 0.001), D-dimers (*p* = 0.033), fibrinogen (*p* = 0.001), ferritin (*p* = 0.002), CD4/CD8 ratio (*p* < 0.001), and reduced absolute counts of CD3+ T-cells (*p* < 0.001) and CD3negCD16+CD56+ NK-cells (*p* < 0.001) were predictive of an indeterminate result at the QTF-Plus assay (Table 5). At the multivariable logistic regression, total CD3+ absolute counts (*p* = 0.004) and CD4/CD8 ratio (*p* < 0.001) were independent predictors of QFT-Plus indeterminate responses (Table 5). Overall, the reduction in CD3+ T-cell absolute counts and the imbalance of the CD4/CD8 ratio, with a prevalence of CD4+ over CD8+ T cells, predicted an indeterminate result at the QFT-Plus assay.

## 4. Discussion

The main findings of this work have been the evidence of increased rate of indeterminate QFT-Plus assay in COVID-19 patients, with higher percentages observed in patients with a worse outcome and more severe disease. Indeterminate results of the QFT-Plus assay were due to an impaired IFN-γ production upon PHA stimulation. Furthermore, IFN-γ levels assessed in the mitogen tube of the QFT-Plus assay were directly correlated with the absolute count of CD3+ T-lymphocytes and inversely correlated with the CD4/CD8 ratio. 

In this study, the prevalence of indeterminate QFT-Plus assay in COVID-19 hospitalized patients (93/420, 22%) was higher than it would be expected if compared to the general population, according to previous studies involving both immune competent and immunocompromised patients [31,32,33]. Conversely, compared to other publications on COVID-19 subjects, the rate of indeterminate QFT-Plus results reported in our work is slightly inferior (22% compared to 35–65%) [16,17,21,22]. A possible explanation could be the different selection criteria for the execution of the QFT-Plus assay. In other studies, only a limited number of patients were tested, with possible selection biases [16,17,22], while in our study all the patients hospitalized because of COVID-19 were tested with the QFT-Plus assay on hospital admission, for the diagnosis of latent tuberculosis infection. This systematic approach, reduced the selection biases, allowing us to analyze a large cohort of patients (420 subject with a QFT-Plus assay performed on hospital admission), with a good representation of all the degree of severity observed in COVID-19 hospitalized patients. The median time between the onset of symptoms and the blood sampling for the QFT-Plus assay was 10 days (IQR 7–14), in line with previous studies [16].

Interestingly, the results of the present study showed an association between indeterminate QFT-Plus assay, assessed early during hospitalization, and COVID-19 severity, defined by the maximal oxygen support required during the entire hospitalization. Moreover, we confirmed the association of a QFT-Plus indeterminate result with in hospital-mortality [16,22]. Furthermore, an association has emerged between indeterminate QFT-Plus results, age > 65 years and male sex, which are well known risk factors for severe forms of COVID-19 [3]. A higher prevalence of indeterminate QFT-Plus assay in our series was observed in patients who needed intensive care, due to a greater COVID-19 severity thus evidencing a limitation of the QFT-Plus assay for the identification of latent tuberculosis in acute COVID-19 critically ill patients.

While other studies showed an association between indeterminate QFT-Plus assay and immunosuppressive treatment with monoclonal antibodies against IL-6-receptor [16] and corticosteroids [16,33,34,35], in our series we could not identify such a statistical association. This could be due to different aspects: (1) QFT-Plus assay was generally performed before steroid and anti-IL-6 receptor administration; (2) a limited number of patients included in the analysis received IL-6 receptor inhibitors (<10%, 41/420). After removing from the analysis all the patients who received steroids and IL-6-receptor inhibitors before the blood sampling for QFT-Plus assay, an increased rate of indeterminate results was confirmed in patients with more severe disease and poorer outcome, as shown by other authors [22]. It should be considered that the QFT-Plus assay was generally performed on admission to the Infectious Disease Ward, before patients underwent steroid or IL-6 receptor inhibitors, although in many cases a previous low dose home steroid treatment cannot be excluded with certainty.

It is known that the QFT-Plus assay results are influenced by a dysregulated immune system, such as in patients with inflammatory bowel disease on treatment with corticosteroid and/or TNF-α inhibitors [34,35] and in HIV-positive patients [36,37].

Moreover, several studies have shown a link between indeterminate QFT-Plus assay results and lymphopenia [38,39]. Lymphopenia often characterizes the most severe forms of SARS-CoV-2 infection [4,26,27] supporting the hypothesis of a direct correlation with increased rate of indeterminate QFT-Plus assay [16,17,21]. Considering that peripheral blood lymphocyte subsets were studied in almost all the patients included in the analysis, we could demonstrate that patients with indeterminate QFT-Plus assay had reduced circulating numbers of total CD3+, CD3+CD4+, CD3+CD8+ T-lymphocytes, and CD3negCD16+CD56+ NK-cells. No differences were observed for B-lymphocytes. Interestingly, patients with indeterminate results had also an increased CD4/CD8 ratio. Furthermore, total T-lymphocyte, CD4 and CD8 subset absolute counts were directly correlated to the IFN-γ production after PHA stimulation. 

In some studies, the increased rate of QFT-Plus indeterminate results was attributed to the “cytokine storm” associated with the severe forms of COVID-19 [22]. Our results do not support this hypothesis, for several reasons. Although inflammation markers (CRP, D-dimers, fibrinogen, ferritin) were predictive of and found increased in patients with indeterminate QFT-Plus at the univariable analysis, only CD3+ T-cells count and CD4/CD8 ratio remained as independent predictors of an indeterminate result of the QFT-Plus assay, at the multivariable analysis. Moreover, IL-6 levels did not significantly differ in COVID-19 patients with a determinate or indeterminate QFT-Plus assay. TNF-α levels were even reduced in patients with an indeterminate result at the QFT-Plus assay compared to patients with a determinate result. Finally, in all the cases of indeterminate results, in the Nil tube (representing the basal unstimulated IFN-γ production) the levels of IFN-γ were not increased, while in the Mitogen-Nil condition the IFN-γ levels were below 0.5 IU/mL, demonstrating an impairment in the production of this cytokine, rather than a dysregulated hyperproduction. Taken together all these elements lead to the conclusion that in our cohort the increased rate of indeterminate QFT-Plus results depends on the reduction in T-lymphocyte absolute counts found in the most severe cases of SARS-CoV-2 infection, with an imbalance of CD4/CD8 ratio due to a prevalence of CD4 and reduction in CD8 T-cells. Nevertheless, we cannot completely rule out a possible role played by lymphocyte dysfunction. 

Besides the reduction in circulating T-lymphocyte absolute counts, some authors demonstrated the presence of phenotypic and functional abnormalities in COVID-19 patients [11]. De Biasi et al. showed several alterations involving naïve, central memory, effector memory, and terminally differentiated T-cells, as well as regulatory T-cells and PD1+CD57+ exhausted T-cells [40]. An impairment in T-lymphocyte function, with reduced capacity to produce IFN-γ and TNF-α upon stimulation with anti-CD3/anti-CD28 monoclonal antibody has been evidenced in COVID-19 patients. Interestingly, IL-17 seemed to restore in vitro IFN-γ production by T-lymphocyte of COVID-19 patients [4]. Another consideration is the fact that in HIV patients with CD4 cell count below 100 cell/µL, the rate of indeterminate results of the QuantiFERON-TB Gold test is inferior to the value we and other groups have observed in COVID-19 patients [36], thus indicating the concomitant contribution of T-lymphocyte dysregulation together with absolute count reduction.

Our results demonstrated a reduced production of IFN-γ by T-lymphocytes upon PHA stimulation in COVID-19 patients with a more severe disease. Furthermore, a negative correlation between inflammatory markers (such as CRP, ferritin, D-dimers) and IFN-γ production by peripheral blood T-lymphocyte was also identified. These results seem to contrast with the evidence obtained by Karki et al., who demonstrated that the combination of IFN-γ and TNF-α was able to recapitulate the cellular and tissue damages observed in severe COVID-19 patients with acute respiratory distress syndrome (ARDS), both in vitro cultures and in vivo animal models. Moreover, IFN-γ and TNF-α blocking was able to increase the survival of SARS-CoV-2 infected mice [41]. T-cell dysregulation during COVID-19 disease is characterized by complex and heterogeneous series of events, which encompass both T-cell hyperactivation and exhaustion [42]. Moreover, T-cell count reduction and impaired IFN-γ production in peripheral blood can reflect activated T-cell margination, recruitment and sequestration in lung tissue and lymphoid organs with local hyperinflammation and secretion of inflammatory cytokines [42]. Considering this complexity, more studies comparing T-cell number, phenotype and function in both peripheral blood and lung tissue are needed, to clarify the role of these immune cells in the pathogenesis of COVID-19. 

The strengths of this study are represented by the large number of COVID-19 patients included and the systematic approach of performing the QFT-Plus assay on hospital admission, avoiding selection biases.

Some limitations are represented by the retrospective design and the cross-sectional assessment of the QFT-Plus assay. It would be useful to repeat the QFT-Plus after the acute phase of the disease, in patients who survived COVID-19 and normalized peripheral blood lymphocyte counts. Being a monocentric study, our results need to be confirmed and verified in other settings with different cohort of COVID-19 patients.

## 5. Conclusions

In conclusion, our study confirmed an increased prevalence of indeterminate QFT-Plus assay in a large cohort of patients hospitalized because of SARS-CoV-2 infection, and demonstrated a direct link between the impaired IFN-γ production in the Mitogen-Nil condition and the reduction in peripheral blood T-lymphocytes in COVID-19 patients.

## Figures and Tables

**Figure 1 life-12-00244-f001:**
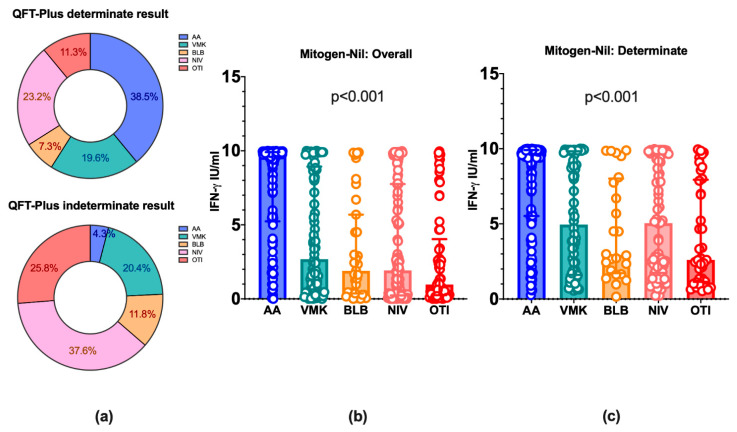
QFT-Plus indeterminate results and IFN-γ production in COVID-19 patients stratified according to the maximal oxygen supply needed during hospitalization. (**a**) Rates of determinate and indeterminate results at the QFT-Plus assay in COVID-19 patients stratified according to the maximal oxygen support needed during the entire hospitalization are represented. (**b**) The levels of IFN-γ in the Mitogen-Nil condition of the QFT-Plus assay in COVID-19 patients stratified according to the maximal oxygen support received during the entire hospitalization are represented. (**c**) As in (**b**), after removing COVID-19 patients with an indeterminate result of the QFT-Plus Assay. The histogram height represents the median value. Whiskers represent the interquartile range. QFT-Plus: QuantiFERON-TB Gold Plus; IFN-γ: interferon-γ; AA: ambient air; VMK: Venturi oxygen mask; NRM: Non-Rebreather Mask; NIV: non-invasive ventilation; OTI: orotracheal intubation for mechanical ventilation.

**Figure 2 life-12-00244-f002:**
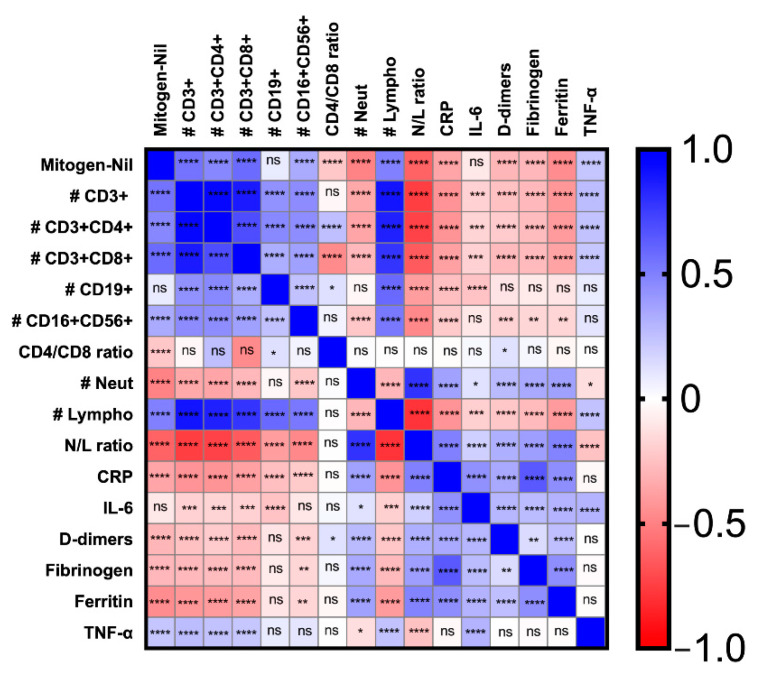
Correlation between IFN-γ levels in the Mitogen-Nil condition of the QFT-Plus assay and laboratory parameters in COVID-19 patients. The statistical significance is represented for each couple of parameters by asterisks. The color intensity in each square box represents the value of the Spearman rho coefficient, ranging from −1 (red), through 0 (white) to 1 (blue). QFT-Plus: QuantiFERON-TB Gold Plus; IFN-γ: interferon-γ; #: absolute counts; Neut: neutrophils; Lympho: lymphocyte; N/L ratio: neutrophils-to-lymphocytes ratio; CRP: C-reactive protein; IL-6: interleukine-6; TNF-α: tumor necrosis factor-α. 0.01 < * < 0.05; 0.001 < ** < 0.01; 0.0001 < *** < 0.001; **** < 0.0001.

**Table 1 life-12-00244-t001:** Demographics and clinical characteristic of the cohort, overall and after stratification for outcome (survivors vs. non-survivors) and COVID-19 severity (non-severe vs. severe).

	All Patients (N = 420)	Survivors (76.9%)	Non-Survivors (23.1%)	*p* *	Non-Severe (50.7%)	Severe (49.3%)	*p* **
Male/Female (%)	66.4/33.6	63.2/36.8	77.3/22.7	0.010	58.7/41.3	74.4/25.6	<0.001
Age (years)	65.0 (52.0–77.0)	61.0 (49.0–72.5)	73.0 (68.0–82.0)	<0.001	59.0 (47.0–75.0)	69.0 (57.5–78.0)	<0.001
Age > 65 years (%)	49.0	39.6	80.4	<0.001	38.5	60.0	<0.001
Charlson Index	4.0 (2.0–5.0)	3.0 (2.0–5.0)	6.0 (4.0–7.0)	<0.001	3.0 (1.0–5.0)	4.0 (3.0–6.0)	<0.001
Cardiovascular (%)	56.4	50.8	75.3	<0.001	49.8	63.3	0.005
Diabetes (%)	22.1	19.5	30.9	0.017	20.7	23.7	0.457
Obesity (%)	19.8	20.4	17.5	0.528	16.4	23.2	0.082
Pulmonary (%)	14.3	10.2	27.8	<0.001	10.8	17.9	0.038
Neurological/ Psychiatric (%)	13.8	10.5%	24.7	<0.001	13.1	14.5	0.689
Solid tumor (%)	13.6	11.5	20.6	0.021	10.8	16.4	0.092
Endocrinological (%)	9.5	10.2	7.2	0.377	9.9	9.2	0.812
Renal (%)	9.3	5.3	22.7	<0.001	8.5	10.1	0.550
Cerebrovascular (%)	8.1	5.3	17.5	<0.001	6.1	10.1	0.129
Hematological (%)	6.2	4.0	13.4	<0.001	2.8	9.7	0.004
Immunological/ Rheumatological (%)	5.2	5.9	3.1	0.280	8.0	2.4	0.010
Viral Hepatitis (%)	1.9	1.9	2.1	0.897	1.9	1.9	0.967
Other comorbidities (%)	24.3	23.5	26.8	0.510	23.5	25.1	0.694
ICU (%)	19.8	8.4	57.7	<0.001	2.8	37.2	<0.001
Steroid treatment (%)	62.4	56.0	83.5	<0.001	34.7	90.8	<0.001
Anti-IL6R (%)	9.8	9.6	10.3	0.846	4.7	15.0	<0.001
ΔT Symp-NPhS (days)	5.0 (1.0–8.0)	5.0 (2.0–8.0)	4.0 (0.0–8.0)	0.102	4.0 (1.0–7.0)	5.0 (2.0–8.0)	0.237
ΔT NPhS-QFTP (days)	4.0 (2.0–7.0)	4.0 (2.0–7.0)	3.0 (2.0–7.0)	0.467	4.0 (2.0–7.0)	4.0 (2.0–8.0)	0.890
ΔT Symp-QFTP (days)	10.0 (7.0–14.0)	10.0 (7.0–14.0)	9.0 (5.0–12.8)	0.051	10.0 (6.0–14.0)	10.0 (7.0–14.0)	0.497
ΔT Symp-TBNK (days)	8.5 (5.0–12.0)	9.0 (5.0–12.0)	7.0 (4.0–11.0)	0.036	8.0 (4.0–11.0)	9.0 (5.0–12.0)	0.045
ΔT TBNK-QFTP (days)	0.0 (0.0–3.0)	0.0 (0.0–3.0)	0.0 (0.0–3.0)	0.853	1.0 (0.0–3.0)	0.0 (0.0–3.0)	0.010

Quantitative variables are presented as median (interquartile range); categorical variables are presented as percentages. * comparison between survivors vs. non-survivors; ** comparison between non-severe vs. severe COVID-19 patients. Cardiovascular comorbidities included heart failure, coronary artery disease, cardiomyopathies and hypertension; diabetes included both type I and II diabetes mellitus; obesity was defined as a body mass index ≥ 30 kg/m^2^; pulmonary comorbidities included all kind of chronic lung diseases; neurological/psychiatric comorbidities consisted of all chronic neurological conditions, including dementia, as well as mental health disorders and depression; solid tumor included all malignant neoplastic diseases; endocrinological comorbidities included non-neoplastic endocrinological disorders; renal comorbidities included chronic kidney disease; cerebrovascular comorbidities included stenosis, thrombosis, embolism and hemorrhages; hematological comorbidities included malignancies, red blood cell disorders, platelet disorders; immunological/rheumatological disorders included autoimmune and connective tissue diseases; viral hepatitis included active or past HBV and/or HCV infection; other comorbidities included clinical relevant conditions not included in the above mentioned conditions. ICU: Intensive care unit; anti-IL6R: anti-interleukine-6 receptor; ΔT Symp-NPhS: days from symptoms’ onset to the first positive nasopharyngeal swab for SARS-CoV-2 RNA detection with RT-PCR; ΔT NPhS-QFTP: days from the first positive nasopharyngeal swab for SARS-CoV-2 RNA detection with RT-PCR to QuantiFERON-TB Gold Plus sampling day; ΔT Symp-QFTP: days from symptoms’ onset to QuantiFERON-TB Gold Plus sampling day; ΔT Symp-TBNK: days from symptoms’ onset to peripheral blood T-, B-, NK-lymphocyte assessment day; ΔT TBNK-QFTP: days from peripheral blood T-, B-, NK-lymphocyte assessment to QuantiFERON-TB Gold Plus sampling day.

**Table 2 life-12-00244-t002:** Laboratory parameters at Infectious diseases ward admission of the study cohort, overall and after stratification for outcome (survivors vs. non-survivors) and COVID-19 severity (non-severe vs. severe).

	All patients	Survivors	Non-Survivors	*p* *	Non-Severe	Severe	*p* **
Neutrophils (cells/µL)	4740 (3170–7470)	4475 (2910–6675)	6300 (3845–9005)	<0.001	3810 (2480–5710)	6365 (3990–8930)	<0.001
Lymphocytes (cells/µL)	990 (620–1400)	1110 (730–1492)	630 (405–840)	<0.001	1240 (900–1650)	740 (490–1018)	<0.001
N/L ratio	5.1 (2.4–10.1)	4.1 (2.0–7.9)	11.4 (5.5–17.7)	<0.001	3.2 (1.7–5.2)	8.4 (5.0–14.6)	<0.001
CRP (mg/L)	60.1 (25.1–113.5)	44.7 (18.2–98.4)	112.2 (67.3–150.7)	<0.001	36.7 (11.7–76.1)	87.5 (45.3–138.5)	<0.001
IL-6 (pg/mL)	17.4 (7.6–43.4)	13.7 (6.2–31.5)	40.2 (16.5–65.7)	<0.001	12.7 (5.6–29.0)	21.9 (11.5–50.9)	<0.001
TNF-α (pg/mL)	12.8 (7.1–25.4)	12.7 (7.2–26.6)	13.3 (7.0–23.9)	0.833	15.2 (9.0–32.0)	11.1 (6.7–19.1)	<0.001
D-dimer (ng/mL)	942 (519–1601)	807.0 (484.5–1494.0)	1354.5 (924.8–2155.8)	<0.001	800.5 (445.8–1466.8)	1094.0 (655.0–1850.0)	<0.001
Fibrinogen (mg/dL)	550 (418–684)	530.5 (405.0–669.8)	610.0 (447.3–731.0)	0.020	472.0 (364.5–620.0)	625.0 (500.0–736.0)	<0.001
Ferritin (mg/dL)	635 (282–1252)	494.0 (228.5–1064.0)	1156.5 (684.5–1869.0)	<0.001	354.0 (175.0–930.0)	900.0 (450.3–1773.3)	<0.001

Quantitative variables are presented as median (interquartile range). * comparison between survivors vs. non-survivors; ** comparison between non-severe vs. severe COVID-19 patients. Reference values: CRP (mg/L): 0–5.00; IL-6 (pg/mL): <50; TNF-alpha (pg/mL): <12.4; D-dimers (ng/mL): 0–500.00; Fibrinogen (mg/dL): 200.00–400.00; Ferritin: 4.63–204.00. N/L ratio: neutrophils-to-lymphocytes ratio; CRP: C-reactive protein; IL-6: interleukine-6, TNF-α: tumor necrosis factor-α.

**Table 3 life-12-00244-t003:** QFT-Plus results of the study cohort, overall and after stratification for outcome (survivors vs. non-survivors) and COVID-19 severity (non-severe vs. severe).

	All Patients (N = 420)	Survivors (N = 323)	Non-Survivors (N = 97)	*p* *	Non-Severe (N = 213)	Severe (N = 207)	*p* **
Pos/Neg/Indet (%)	8.3/69.6/22.1	9.6/71.2/19.2	4.1/63.9/32.0%	0.013	10.3/78.9/10.8	6.3/59.9/33.8	<0.001
Det/Indet (%)	77.9/ 22.1	80.8/19.2	68.0/32.0	0.008	89.2/10.8	66.2/33.8	<0.001
TB1 (IU/mL)	0.106 (0.055–0.249)	0.112 (0.062–0.302)	0.086 (0.044–0.164)	0.006	0.121 (0.060–0.319)	0.103 (0.050–0.192)	0.027
TB1-Nil (IU/mL)	0.006 (0.000–0.043)	0.007 (0.000–0.052)	0.002 (0.000–0.020)	0.022	0.007 (0.000–0.047)	0.005 (0.000–0.032)	0.078
TB2 (IU/mL)	0.103 (0.057–0.246)	0.109 (0.065–0.271)	0.082 (0.045–0.169)	<0.001	0.111 (0.067–0.289)	0.093 (0.050–0.197)	0.005
TB2-Nil (IU/mL)	0.005 (0.000–0.042)	0.007 (0.000–0.051	0.001 (0.000–0.029)	0.027	0.009 (0.000–0.047)	0.003 (0.000–0.039)	0.067
Mitogen (IU/mL)	3.920 (0.980–10.000)	6.000 (1.285–10.000)	1.430 (0.480–3.550)	<0.001	9.770 (2.100–10.000)	1.910 (0.450–6.670)	<0.001
Mitogen-Nil (IU/mL)	3.677 (0.663–9.657)	5.503 (1.032–9.808)	1.337 (0.273–3.327)	<0.001	8.402 (1.797–9.887)	1.658 (0.272–6.316)	<0.001

Quantitative variables are represented as median (interquartile range); categorical variables are presented as percentages. * comparison between survivors vs. non-survivors; ** comparison between non-severe vs. severe COVID-19 patients. QFT-Plus: QuantiFERON-TB Gold Plus; Pos: positive; Neg: negative; Indet: indeterminate; Det: determinate.

**Table 4 life-12-00244-t004:** Demographic, clinical and laboratory parameters in COVID-19 patients with a determinate or indeterminate result at the QFT-Plus assay.

	All Patients	QFT-Plus Determinate	QFT-Plus Indeterminate	*p*
Male/Female (%)	66.4/33.6	63.9/36.1	75.3/24.7	0.041
Age (years)	65.0 (52.0–77.0)	64 (50–76)	70 (57–79)	0.003
Age > 65 years (%)	49.0	45.6	61.29	0.007
Charlson Index	4.0 (2.0–5.0)	4.0 (2.0–5.0)	4.0 (3.0–6.0)	0.014
ΔT NPhS-QFTP (days)	5.0 (1.0–8.0)	4 (2–7)	5 (2–8)	0.330
ΔT Symp-QFTP (days)	4.0 (2.0–7.0)	10 (6–13)	11 (8–15)	0.086
ΔT TBNK-QFTP (days)	8.5 (5.0–12.0)	0 (0–3)	0 (0–3)	0.355
ICU (%)	19.8%	16.5%	31.2	0.002
Neutrophils (cells/µL)	4740 (3170–7470)	4245 (2835–6685)	6660 (5360–9880)	<0.001
Lymphocytes (cells/µL)	990 (620–1400)	1065 (682–1490)	680 (440–1000)	<0.001
N/L ratio	5.1 (2.4–10.1)	4.0 (2.0–8.0)	10.1 (5.8–18.5)	<0.001
CRP (mg/L)	60.1 (25.1–113.5)	50.6 (20.1–105.6)	101.6 (43.9–149.1)	<0.001
IL-6 (pg/mL)	17.4 (7.6–43.4)	15.7 (7.4–42.8)	18.4 (9.9–47.5)	0.255
TNF-alpha (pg/mL)	12.8 (7.1–25.4)	14.8 (7.7–27.7)	10.2 (5.9–16.9)	<0.001
D-dimer (ng/mL)	942.0 (519.0–1601.0)	839.0 (484.5–1518.5)	1183.5 (738.5–2043.8)	<0.001
Fibrinogen (mg/dL)	550.0 (418.0–684.0)	521.0 (398.0–661.0)	629.0 (487.5–729.0)	<0.001
Ferritin (mg/dL)	635.0 (282.0–1252.0)	459.5 (224.0–1036.8)	1111.0 (751.0–1783.0)	<0.001
# CD3+ (cells/µL)	633.0 (383.0–1025.0)	725.0 (452.3–1096.8)	376.5 (242.0–615.5)	<0.001
# CD3+CD4+ (cells/µL)	386.0 (232.0–632.0)	433.0 (268.3–708.0)	247.5 (151.5–440.0)	<0.001
# CD3+CD8+ (cells/µL)	196.0 (116.0–357.5)	240.5 (140.8–402.3)	104.5 (74.5–172.5)	<0.001
CD4/CD8 ratio	1.9 (1.2–2.8)	1.82 (1.15–2.50)	2.43 (1.77–3.48)	<0.001
# CD19+ (cells/µL)	106.5 (62.0–161.8)	106.5 (62.0–161.3)	106.5 (63.5–171.5)	0.882
# CD3negCD16+CD56+ (cells/µL)	129.0 (80.0–214.3)	146.5 (91.5–221.3)	93.5 (57.0–158.8)	<0.001

Quantitative variables are presented as median (interquartile range); qualitative variables are presented as percentages. Reference values: CRP (mg/L): 0–5.00; IL-6 (pg/mL): <50; TNF-alpha (pg/mL): <12.4; D-dimers (ng/mL): 0–500.00; Fibrinogen (mg/dL): 200.00–400.00; Ferritin: 4.63-204.00, CD3+(%): 55–84; CD3+(cells/μL): 690–2540; CD3+CD4+(cells/μL): 410–1590; CD3+CD8+(cells/μL): 190–1140; CD19+(cells/μL): 90–660; CD3negCD16+CD56+(cells/μL): 90–590; CD4/CD8 ratio: 1.5–2.5. QFT-Plus: QuantiFERON-TB Gold Plus; ΔT NPhS-QFTP: days from the first positive nasopharyngeal swab for SARS-CoV-2 RNA detection with RT-PCR to QuantiFERON-TB Gold Plus sampling day; ΔT Symp-QFTP: days from symptoms’ onset to QuantiFERON-TB Gold Plus sampling day; ΔT TBNK-QFTP: days from peripheral blood T-, B-, NK-lymphocyte assessment to QuantiFERON-TB Gold Plus sampling day; N/L ratio: neutrophils-to-lymphocytes ratio; CRP: C-reactive protein; IL-6: interleukine-6, TNF-α: tumor necrosis factor-α; #: absolute count.

**Table 5 life-12-00244-t005:** Univariable and multivariable logistic regression analysis: factors associated with an indeterminate response of the QTF-Plus assay in COVID-19 patients.

	Univariable				Multivariable			
	Odds Ratio	95% CI		*p*	Odds Ratio	95% CI		*p*
Sex (M)	1.718	1.019	2.897	**0.042**	1.120	0.514	2.407	0.788
Age > 65 years	1.891	1.181	3.028	**0.008**	0.877	0.404	1.901	0.739
Charlson index	1.110	1.015	1.213	**0.022**	1.077	0.904	1.283	0.405
Tocilizumab	1.416	0.631	3.176	0.398	/	/	/	/
Sarilumab	1.783	0.437	7.272	0.420	/	/	/	/
Anti-IL-6R before QFT-Plus	2.611	0.965	7.062	0.059	/	/	/	/
Steroid before QFT-Plus	0.570	0.164	1.980	0.377	/	/	/	/
N/L Ratio	1.100	1.066	1.135	**<0.001**	1.027	0.982	1.074	0.240
CRP	1.007	1.003	1.010	**<0.001**	0.999	0.992	1.005	0.633
IL-6	1.000	0.999	1.001	0.816	/	/	/	/
D-dimers	1.000	1.000	1.000	**0.033**	1.000	1.000	1.000	0.657
Fibrinogen	1.002	1.001	1.003	**0.001**	1.000	0.998	1.003	0.867
Ferritin	1.000	1.000	1.000	**0.002**	1.000	1.000	1.000	0.303
# CD3+	0.997	0.996	0.998	**<0.001**	0.998	0.996	0.999	**0.005**
CD4/CD8 ratio	1.303	1.109	1.531	**0.001**	1.546	1.216	1.965	**<0.001**
# CD19+	1.000	0.999	1.000	0.677	/	/	/	/
# CD3negCD16+CD56+	0.996	0.994	0.999	**0.002**	1.000	0.997	1.003	0.857

QFT-Plus: QuantiFERON-TB Gold Plus; 95% CI: 95% confidence interval; N/L ratio: neutrophils-to-lymphocytes ratio; CRP: C-reactive protein; IL-6: interleukine-6; #: absolute counts.

## Data Availability

The data presented in this study are available on request from the corresponding author. The data are not publicly available due to privacy and ethical concerns.

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
