# Peer review of "In Patients with Severe COVID-19, the Profound Decrease in the Peripheral Blood T-Cell Subsets Is Correlated with an Increase of QuantiFERON-TB Gold Plus Indeterminate Rates and Reflecting a Reduced Interferon-Gamma Production"

_life, 2022, doi:10.3390/life12020244_

Round 1

Reviewer 1 Report

The aim of this retrospective study was to determine the prevalence of indeterminate QFT-Plus test in hospitalized patients with COVID-19, to analyze its relationship with severity in-hospital mortality, and to identify the factors influencing the indeterminate results. CD3+ T-cell absolute counts and CD4/CD8 ratio were independent predictors of indeterminate results at the QFT-Plus. Increased rate of indeterminate QFT-Plus was found in non-survivors and in severe COVID-19 patients.

C. Materials and methods

Line 136. Categorical variables are presented as absolute frequency and (percentages), while quantitative variables are presented as medians and interquartile ranges (IQR).

I'm not sure that the word “percentage” should be inside the brackets. It might be more appropriate to include the abbreviation (%) or delete the brackets. In this way, in tables these definitions are more homogeneous but you might use always the same kind of brackets: () or [].

D. Results

General consideration:

Sections 3.1 (Study population) and 3.2 (Laboratory parameters and inflammation markers) contain more than 50 lines and 2 tables. It is not an explicit aim of your study and also, you do not discuss this information later. I suggest you consider whether some parts of the text / tables could be dispensable.

Line 150: Four patients were excluded from the analysis because the QFT-Plus assay was  performed more than 30 days after the first positive SARS-CoV-2 NPhS, therefore 420 patients were evaluated

As far as I know, a QFT-Plus is not recommended for all hospitalized COVID-19 patients but it was done routinely at your hospital. Furthermore, you later explain that lymphocyte subsets were performed in almost all patients. Maybe it would be necessary to clarify.

Line 156-162: Most of the enrolled subjects had at least one underlying disease. Specifically: 237 patients (56.4%) had concomitant cardiovascular diseases, mainly hypertension; 93 (22.1%) had diabetes mellitus; 83 were obese (19.8%); ... 34 (8.1%) had cerebrovascular disease;

I don’t understand why you included hypertension within cardiovascular disease and not other risk factors like diabetes. It may be necessary to reclassify in a more academic way or to expand the section of definitions specifying which conditions are included into each variable.

Line 162: Based on the sum of comorbidities presented by each patient, a Comorbidity Score was obtained, resulting in a median value of 2 (IQR 1-3) for the whole cohort.

Why didn't you use a validated scale (e.g. Charlson) letting other centers reproduce your results? Taking this into account, perhaps it would be better highlight which comorbidities (eg cardiovascular, pulmonary, hematological, etc.) are associated with higher mortality / severity (lines 165-170).

Tables: For quantitative data, differences were assessed using the non-parametric Mann-Whitney test; for qualitative data differences were assessed using the 2-tailed Chi2 test.

You are repeating the methods specified in the Statistical Analysis section, so consider if you need to repeat it at all tables again.

I do not understand why you use different types of brackets depending on whether it is a (percentage) or [IQR].

Line 206: CRP, IL-6, D-dimers, fibrinogen and ferritin were significantly increased in non-survivors compared to survivors (p<0.001, p<0.001, p<0.001, p<0.02, p<0.001, respectively) and in severe compared to non-severe patients (p<0.001, p<0.001, p<0.001, p<0.02, p<0.001, respectively).

You already commented that there are significant differences and in the tables you show the p-values, median and IQR. It might be enough if you provide the information only once. This could be applicable for other parts of the text.

Lines 210-211: No statistical or clinically significant differences were observed for TNF-α, after comparing survivors vs non-survivors, or severe vs non-severe COVID-19 patients (Table 2).

There are significant differences for TNF-a between severe vs non-severe COVID-19 patients (p<0.001) in the table 2. Fuerthermore, why do you use an asterisk only for TNF-a in the table 2?

Lines 231-234 and 242-244:

They are repeated.

Line 241-242: The Mitogen and Mitogen-Nil conditions in the QFT-P assay are a measure of the 241 nonspecific T-cell stimulation and an indicator of the immune dysregulation.

This sentence is more appropriate for introduction or discussion.

Line 345-346 At the multivariable logistic regression, after including in the model all the factors with significant association at the univariable analysis, ... and Line 355-356: All the parameters significantly associated with an indeterminate QFT-Plus assay were included in the multivariable analysis.

Statistical Analysis section could be the better place to specify how you performed the multivariable analysis.

E. Discussion

General consideration:

Discussion should start with a summary of the main findings.

4th to 6th paragraphs are a little bit repetitive and maybe you might consider rewrite some parts.

Line 408: Moreover, considering that peripheral blood lymphocyte subsets were studied in almost all the patients included in the analysis, …

In table 4 it seems that blood lymphocyte subsets were performed in all patients. if not, please correct the text or the table.

Line 437: Nevertheless, we cannot completely rule out a possible role played by lymphocyte dysfunction.

Reading the last paragraphs of the discussion, it could be inferred that lymphopenia (and especially T subsets) is the only factor contributing to an indeterminate QFT-Plus. You should consider other studies that question lymphopenia as the only factor. Some of them are in your references:   

-Your reference 4 (Remy et al) that showed an impaired IFN-É£ production in standardized density PBMCs per well using ELISpot.

-Your reference 35 (Santin et al) that showed 9.5% of indeterminate results in VIH-positive patient with CD4+ cell counts lower than 100. Although this group of HIV has an important lymphopenia, the percentage of indeterminate patients is lower than patients with COVID-19.

A key to explain it could be your reference 11 (Vabret et al) that explain how CoV suppresses IFN release.

Line 439-442 (limitations): You performed an unicenter study, could this fact make the results not extrapolable to other populations? Maybe it would be good to talk about the strengths of your study too.

F. Conclusions

Line 445-447: ... and demonstrated a direct link between the impaired IFN-γ production in the Mitogen-Nil condition and the reduction in peripheral blood T-lymphocytes in severe COVID-19 patients

Your results are not limited to patients with severe COVID-19 so it may be best to rewrite it as "and the reduction in peripheral blood T-lymphocytes in COVID-19 patients”.

Author Response

Replies to Reviewer 1

The aim of this retrospective study was to determine the prevalence of indeterminate QFT-Plus test in hospitalized patients with COVID-19, to analyze its relationship with severity in-hospital mortality, and to identify the factors influencing the indeterminate results. CD3+ T-cell absolute counts and CD4/CD8 ratio were independent predictors of indeterminate results at the QFT-Plus. Increased rate of indeterminate QFT-Plus was found in non-survivors and in severe COVID-19 patients.

  1. Materials and methods

Line 136. Categorical variables are presented as absolute frequency and (percentages), while quantitative variables are presented as medians and interquartile ranges (IQR).

I'm not sure that the word “percentage” should be inside the brackets. It might be more appropriate to include the abbreviation (%) or delete the brackets. In this way, in tables these definitions are more homogeneous but you might use always the same kind of brackets: () or [].

We have removed the word “percentage” and included the abbreviation “(%)”. We used the same kind of brackets throughout the manuscript: ().

  1. Results

General consideration:

Sections 3.1 (Study population) and 3.2 (Laboratory parameters and inflammation markers) contain more than 50 lines and 2 tables. It is not an explicit aim of your study and also, you do not discuss this information later. I suggest you consider whether some parts of the text / tables could be dispensable.

We removed the paragraph describing comorbidities, which was a repetition of table 1.

Line 150: Four patients were excluded from the analysis because the QFT-Plus assay was performed more than 30 days after the first positive SARS-CoV-2 NPhS, therefore 420 patients were evaluated

As far as I know, a QFT-Plus is not recommended for all hospitalized COVID-19 patients but it was done routinely at your hospital. Furthermore, you later explain that lymphocyte subsets were performed in almost all patients. Maybe it would be necessary to clarify.

The rationale of performing QFT-Plus assay has been explained in the introduction (lines 48-52).

We added a sentence in the Materials and Method section, explaining this point (lines 88-93):

“In our hospital QTF-Plus assay was routinely performed in all COVID-19 hospitalized patients to identify the presence of latent tuberculosis infection (LTBI). On hospital ad-mission, peripheral blood lymphocyte absolute counts were also routinely assessed in COVID-19 patients. Both tests were used to better characterize COVID-19 patients, con-sidering the possibility of starting a treatment with anti-IL-6 receptor and other im-munomodulatory therapies.”

Line 156-162: Most of the enrolled subjects had at least one underlying disease. Specifically: 237 patients (56.4%) had concomitant cardiovascular diseases, mainly hypertension; 93 (22.1%) had diabetes mellitus; 83 were obese (19.8%); ... 34 (8.1%) had cerebrovascular disease;

I don’t understand why you included hypertension within cardiovascular disease and not other risk factors like diabetes. It may be necessary to reclassify in a more academic way or to expand the section of definitions specifying which conditions are included into each variable.

We included a description of which conditions are included into each variable in the caption of table 1

“Cardiovascular comorbidities included heart failure, coronary artery disease, cardiomyopathies and hypertension; diabetes included both type I and II diabetes mellitus; obesity was defined as a body mass index ≥30 kg/m2; pulmonary comorbidities included all kind of chronic lung diseases; neurological/psychiatric comorbidities consisted of all chronic neurological conditions, including dementia, as well as mental health disorders and depression; solid tumor included all malignant neoplastic diseases; endocrinological comorbidities included non-neoplastic endocrinological disorders; renal comorbidities included chronic kidney disease; cerebrovascular comorbidities included stenosis, thrombosis, embolism and hemorrhages; hematological comorbidities included malignancies, red blood cell disorders, platelet disorders; immunological/rheumatological disorders included autoimmune and connective tissue diseases; viral hepatitis included active or past HBV and/or HCV infection; other comorbidities included clinical relevant conditions not included in the above mentioned conditions.”

Line 162: Based on the sum of comorbidities presented by each patient, a Comorbidity Score was obtained, resulting in a median value of 2 (IQR 1-3) for the whole cohort.

Why didn't you use a validated scale (e.g. Charlson) letting other centers reproduce your results? Taking this into account, perhaps it would be better highlight which comorbidities (eg cardiovascular, pulmonary, hematological, etc.) are associated with higher mortality / severity (lines 165-170).

We calculated the Charlson Comorbidity Index for all the patients and reported the results in table 1. We also replaced the comorbidity index with the Charlson Comorbidity Index throughout the manuscript.

As suggested, we also highlighted the comorbidities associated to a worse outcome (lines 175-180):

“Notably, cardiovascular, pulmonary, neurological/psychiatric, renal, cerebrovascular, hematological disorders, diabetes and solid tumors were associated with an increased mortality (p<0,001, p<0,001, p<0,001, p<0,001, p<0,001, p<0,001, p=0,017 and p=0,021, respectively). Cardiovascular, pulmonary, hematological and immunological/rheumatological disorders were also associated to a more severe disease (p=0,005, p=0,038, p=0,004 and p=0,010, respectively).”

Tables: For quantitative data, differences were assessed using the non-parametric Mann-Whitney test; for qualitative data differences were assessed using the 2-tailed Chi2 test.

You are repeating the methods specified in the Statistical Analysis section, so consider if you need to repeat it at all tables again.

We removed the statistical method from table captions.

I do not understand why you use different types of brackets depending on whether it is a (percentage) or [IQR].

We modified the text and used the same type of brackets throughout the manuscript: ().

Line 206: CRP, IL-6, D-dimers, fibrinogen and ferritin were significantly increased in non-survivors compared to survivors (p<0.001, p<0.001, p<0.001, p<0.02, p<0.001, respectively) and in severe compared to non-severe patients (p<0.001, p<0.001, p<0.001, p<0.02, p<0.001, respectively).

You already commented that there are significant differences and in the tables you show the p-values, median and IQR. It might be enough if you provide the information only once. This could be applicable for other parts of the text.

We agree with the Reviewer and simplified the text as follows (lines 228-230):

“CRP, IL-6, D-dimers, fibrinogen and ferritin were significantly different after comparing survivors with non-survivors and non-severe with severe COVID-19 patients (Table 2).

Lines 210-211: No statistical or clinically significant differences were observed for TNF-α, after comparing survivors vs non-survivors, or severe vs non-severe COVID-19 patients (Table 2).

There are significant differences for TNF-a between severe vs non-severe COVID-19 patients (p<0.001) in the table 2. Furthermore, why do you use an asterisk only for TNF-a in the table 2?

We agree with the reviewer, therefore, the sentence was modified as follows (lines 230-232):

Although TNF-α did not significantly differ after comparing survivors vs non-survivors, its levels were lower in severe than non-severe COVID-19 patients (p<0,001) (Table 2).

The asterisk for TNF-a in table 2 was a typo.

Lines 231-234 and 242-244:

They are repeated.

We removed the paragraph from line 242 to 244.

Line 241-242: The Mitogen and Mitogen-Nil conditions in the QFT-P assay are a measure of the 241 nonspecific T-cell stimulation and an indicator of the immune dysregulation.

This sentence is more appropriate for introduction or discussion.

We moved the sentence in the introduction section (lines 64-66).

“Therefore, the Mitogen (and Mitogen-Nil) condition in the QFT-Plus assay is a measure of the nonspecific T-cell stimulation and an indicator of the immune dysregulation.”

Line 345-346 At the multivariable logistic regression, after including in the model all the factors with significant association at the univariable analysis, ... and Line 355-356: All the parameters significantly associated with an indeterminate QFT-Plus assay were included in the multivariable analysis.

Statistical Analysis section could be the better place to specify how you performed the multivariable analysis.

As suggested, we removed the sentences from the results section and added a statement in the statistical methods (lines 152-155):

“Multivariable regression analysis was used to assess the factors associated to QFT-Plus indeterminate results. With this aim, all the factors significantly associated with an indeterminate QFT-Plus assay at the univariable analysis, were included in the multivariable regression model.”

  1. Discussion

General consideration:

Discussion should start with a summary of the main findings.

We added a small summary of the main findings (366-372).

“The main findings of this work have been the evidence of increased rate of inde-terminate QFT-Plus assay in COVID-19 patients, with higher percentages observed in patients with a worse outcome and more severe disease. Indeterminate results of the QFT-Plus assay were due to an impaired IFN-γ production upon PHA stimulation. Furthermore, IFN-γ levels assessed in the mitogen tube of the QFT-Plus assay were directly correlated with the absolute count of CD3+ T-lymphocytes and inversely corre-lated with the CD4/CD8 ratio.”

4th to 6th paragraphs are a little bit repetitive and maybe you might consider rewrite some parts.

As suggested by the Reviewer we rewrote paragraph 4 to 6 of the Discussion.

Line 408: Moreover, considering that peripheral blood lymphocyte subsets were studied in almost all the patients included in the analysis, …

In table 4 it seems that blood lymphocyte subsets were performed in all patients. if not, please correct the text or the table.

Peripheral blood lymphocyte subsets were systematically assessed in COVID-19 hospitalized patients. For a few patients (18/420) there were technical and logistic issues (clotting, hemolysis, insufficient sample quantity, etc.), therefore the test was not available. We modified table 4 and removed the total number of patients in the columns’ headings. We also added a sentence in section 3.1 to clarify this point (lines 166-167):

“For 18 patients, peripheral blood T-, B-, NK-lymphocyte absolute counts were not available.”

Line 437: Nevertheless, we cannot completely rule out a possible role played by lymphocyte dysfunction.

Reading the last paragraphs of the discussion, it could be inferred that lymphopenia (and especially T subsets) is the only factor contributing to an indeterminate QFT-Plus. You should consider other studies that question lymphopenia as the only factor. Some of them are in your references:   

-Your reference 4 (Remy et al) that showed an impaired IFN-É£ production in standardized density PBMCs per well using ELISpot.

-Your reference 35 (Santin et al) that showed 9.5% of indeterminate results in VIH-positive patient with CD4+ cell counts lower than 100. Although this group of HIV has an important lymphopenia, the percentage of indeterminate patients is lower than patients with COVID-19.

A key to explain it could be your reference 11 (Vabret et al) that explain how CoV suppresses IFN release.

We added a paragraph to the discussion section, in which we underline the possible contribution of T-lymphocyte dysregulation to the increased rate of QFT-Plus indeterminate results (lines 444-456):

“Besides the reduction in circulating T-lymphocyte absolute counts, some authors demonstrated the presence of phenotypic and functional abnormalities in COVID-19 patients [11]. De Biasi et al. showed several alterations involving naïve, central memory, effector memory and terminally differentiated T-cells, as well as regulatory T-cells and PD1+CD57+ exhausted T-cells [38]. An impairment in T-lymphocyte function, with re-duced capacity to produce IFN-γ and TNF-α upon stimulation with anti-CD3/anti-CD28 monoclonal antibody has been evidenced in COVID-19 patients. Interestingly, IL-17 seemed to restore in vitro IFN-γ production by T-lymphocyte of COVID-19 patients [4]. Another consideration is the fact that in HIV patients with CD4 cell count below 100 cell/µl, the rate of indeterminate results of the QuantiFERON®-TB Gold test is inferior to the value we and other groups have observed in COVID-19 patients [34], thus indicating the concomitant contribution of T-lymphocyte dysregulation together with absolute count reduction.”

Line 439-442 (limitations): You performed an unicenter study, could this fact make the results not extrapolable to other populations? Maybe it would be good to talk about the strengths of your study too.

 As suggested by the Reviewer, we added a paragraph to describe the strengths of the study, and added a sentence in the limitation section for the monocentric design of the study (lines 474-476):

“The strengths of this study are represented by the large number of COVID-19 patients included and the systematic approach of performing the QFT-Plus Assay on hospital admission, avoiding selection biases.

Some limitations are represented by the retrospective design and the cross-sectional assessment of the QFT-Plus assay. It would be useful to repeat the QFT-Plus after the acute phase of the disease, in patients who survived COVID-19 and normalized peripheral blood lymphocyte counts. Being a monocentric study, our results need to be confirmed and verified in other settings with different cohort of COVID-19 patients.”

  1. Conclusions

Line 445-447: ... and demonstrated a direct link between the impaired IFN-γ production in the Mitogen-Nil condition and the reduction in peripheral blood T-lymphocytes in severe COVID-19 patients

Your results are not limited to patients with severe COVID-19 so it may be best to rewrite it as "and the reduction in peripheral blood T-lymphocytes in COVID-19 patients”.

We rewrote the sentence following Reviewer’s suggestions

Reviewer 2 Report

In patients with severe COVID-19, the profound decrease in 2 the peripheral blood T-cell subsets is correlated with an in-3 crease of QuantiFERON-TB Gold Plus indeterminate rates and 4 reflecting a reduced interferon-gamma production
The study carried out by the authors of the manuscript is a detailed study of both the effect of the QFT Plus kit and the different immunological and metabolic parameters studied.
I believe that the format and content of Tables 1-3 should be simplified
For example in Table 1 column 1
All patients
(N=420)
Survivors
(N=323; 76.9%)
Non-survivors (N=97; 23.1%) p
Non-severe (N=213; 50.7%)
Severe
(N=207; 49.3%) p
Male/Female
279/141
(66.4%; 33.6%)
204/119
(63.2%; 36.8%)
75/22
(77.3%; 22.7%)
0.010
125/88
(58.7%; 41.3%)
154/53
(74.4%; 25.6%)
<0.001
The Male / Female data does not make much sense to point it out since this data does not correlate in the whole text with any parameter, it would be interesting to separate these two groups and refer to Male and female for the other parameters, if not, it is a data that is not of much interest It can be cited in the Text. The format of the Tables is quite difficult to understand since percentages are shown in parentheses that would not be necessary if the number is shown. If the percentage is chosen, the number can be removed. It is a defect that I see that happens in all the Tables. The Tables would be more understandable.
When speaking of the p value, it must be said between which groups it has been calculated. Survivors vs No survivors; All patients vs Survivors; All patiens vs Non-Non Survivors?. I believe that in this case it refers to Survivors vs Non-survivors but it must be indicated, if you will, in the legend of the Table.
Review the Table 3 format.
I agree with the authors that it would be necessary to carry out this study in patients "It would be useful to repeat the QFT Plus after the acute phase of the disease, in patients who survived COVID-19 and normalized peripheral blood lymphocyte counts." It would be the complement to this article
Two very important references about lymphocyte function in COVID-19 would be missing in this study, I attach them in this review.
Cossarizza et al Cytometry Part A _ 97A: 340–343, 2020
DeBiasi et al NATURE COMMUNICATIONS | (2021) 12:4677

Author Response

Replies to Reviewer 2

In patients with severe COVID-19, the profound decrease in the peripheral blood T-cell subsets is correlated with an increase of QuantiFERON-TB Gold Plus indeterminate rates and reflecting a reduced interferon-gamma production.

The study carried out by the authors of the manuscript is a detailed study of both the effect of the QFT Plus kit and the different immunological and metabolic parameters studied.

I believe that the format and content of Tables 1-3 should be simplified.

For example in Table 1 column 1
All patients
(N=420)
Survivors
(N=323; 76.9%)
Non-survivors (N=97; 23.1%) p
Non-severe (N=213; 50.7%)
Severe
(N=207; 49.3%) p
Male/Female
279/141
(66.4%; 33.6%)
204/119
(63.2%; 36.8%)
75/22
(77.3%; 22.7%)
0.010
125/88
(58.7%; 41.3%)
154/53
(74.4%; 25.6%)
<0.001

The Male / Female data does not make much sense to point it out since this data does not correlate in the whole text with any parameter, it would be interesting to separate these two groups and refer to Male and female for the other parameters, if not, it is a data that is not of much interest It can be cited in the Text.

We thank the Reviewer for the comments. We reported the male/female ratio and percentages in both the text and table 1, as this is a fundamental information for describing the characteristics of the cohort. Furthermore, we showed that the male sex was associated with increased mortality and disease severity (Table 1, p=0.010 and p<0.001, respectively). An increased rate of indeterminate results of the QFT-Plus Assay was also identified in COVID-19 male patients (Table 4, p=0.041). This association was also confirmed at the univariate analysis (p=0.042), although it was not confirmed at the multivariable analysis (Table 5). These analyses summarize what the reviewer suggests, when he/she asks to perform an analysis taking into account male and female patients separately.  

The format of the Tables is quite difficult to understand since percentages are shown in parentheses that would not be necessary if the number is shown. If the percentage is chosen, the number can be removed. It is a defect that I see that happens in all the Tables. The Tables would be more understandable.

As suggested by the Reviewer, we modified all the tables reporting only percentages.   

When speaking of the p value, it must be said between which groups it has been calculated. Survivors vs No survivors; All patients vs Survivors; All patiens vs Non-Non Survivors?. I believe that in this case it refers to Survivors vs Non-survivors but it must be indicated, if you will, in the legend of the Table.
Review the Table 3 format.

As suggested by the reviewer, we specified the groups of patients compared for assessing p values, in table legends (Table1-3):

“ *comparison between survivors vs non-survivors; **comparison between non-severe vs severe COVID-19 patients.”  

I agree with the authors that it would be necessary to carry out this study in patients "It would be useful to repeat the QFT Plus after the acute phase of the disease, in patients who survived COVID-19 and normalized peripheral blood lymphocyte counts." It would be the complement to this article
Two very important references about lymphocyte function in COVID-19 would be missing in this study, I attach them in this review.

Cossarizza et al Cytometry Part A _ 97A: 340–343, 2020

DeBiasi et al NATURE COMMUNICATIONS | (2021) 12:4677

We thank again the reviewer for the comments. We read with attention the suggested papers.

The paper by Cossarizza et al published on Cyotometry Part A in 2020 underlines the need to use adequate approaches, models, and methodologies to characterize the immune system in patients with different stages of COVID-19 disease in order to explore the modifications of innate and adaptive immunity and their relationship with pathogenesis.

The paper by De Biasi et al, published on Nature Communication in 2021 is focused on immunological findings in a cohort of 14 pregnant women with asymptomatic or mild SARS-CoV-2 infection.

After reading these papers, we found another work from the same group, which has been discussed in our manuscript (lines 444-456):

Reference 38.   De Biasi, S.; Meschiari, M.; Gibellini, L.; Bellinazzi, C.; Borella, R.; Fidanza, L.; Gozzi, L.; Iannone, A.; Lo Tartaro, D.; Mattioli, M.; et al. Marked T Cell Activation, Senescence, Exhaustion and Skewing towards TH17 in Patients with COVID-19 Pneumonia. Nat. Commun. 2020, 11, 3434, doi:10.1038/s41467-020-17292-4.

“Besides the reduction in circulating T-lymphocyte absolute counts, some authors demonstrated the presence of phenotypic and functional abnormalities in COVID-19 patients [11]. De Biasi et al. showed several alterations involving naïve, central memory, effector memory and terminally differentiated T-cells, as well as regulatory T-cells and PD1+CD57+ exhausted T-cells [38]. An impairment in T-lymphocyte function, with reduced capacity to produce IFN-γ and TNF-α upon stimulation with anti-CD3/anti-CD28 monoclonal antibody has been evidenced in COVID-19 patients. Interestingly, IL-17 seemed to restore in vitro IFN-γ production by T-lymphocyte of COVID-19 patients [4]. Another consideration is the fact that in HIV patients with CD4 cell count below 100 cell/µl, the rate of indeterminate results of the QuantiFERON®-TB Gold test is inferior to the value we and other groups have observed in COVID-19 patients [34], thus indicating the concomitant contribution of T-lymphocyte dysregulation together with absolute count reduction.” 

Reviewer 3 Report

The authors claim in this manuscript that severe COVID-19 was associated with increased indeterminate QuantiFERON® -TB Gold Plus Assay rates. Moreover, they showed that these rates were associated with peripheral blood T-lymphocyte depletion. 

The number of participants is high, and therefore conclusions could be relevant. Nevertheless, there are some issues in the manuscript.

Major issues:

  • The majority of the results presented here are not novel, and several published papers showed the same results and conclusions
  • Participants are from the first and the second wave. Do the authors determine the viral strains? This could be important for the results' interpretation. 
  • 20 patients were asymptomatic. As the authors said: "Asymptomatic patients were hospitalized in the infectious disease ward because of concomitant clinical conditions that required inpatient care associated with SARS-CoV-2 transmission prevention, although they did not show any sign or symptom related to SARS-CoV-2 infection". These patients with comorbidities but without symptoms could be removed from the analysis, since they can introduce some bias into the study. 
  • Line 211: "No statistical or clinically significant differences were observed for TNF-α, after comparing survivors vs non-survivors, or severe vs non-severe COVID-19 patients (Table 2)". However, in Table 2, the comparison of TNF-a between severe and non-severe was <0.001. Could the authors explain this discrepancy?
  • What is the explanation for the percentage of the QuantiFERON® -TB Gold Plus Assay positivity in survivors vs non-survivors? (Table 3). % of positivity in survivors =9.6% and % of positivity in non-survivors= 4.1%. The same tendency was seen between non-severe (10.3%) and severe (6.3%) COVID-19 patients. It could be excepted that the individuals positive for TB could have more severe symptoms. Indeed, TB is a factor found to influence the severity of the disease. How could the authors explain this point?
  • In Figure 1b: Do the levels of IFN-γ in the Mitogen-Nil condition was shown for all patients or only for those with positive and clear responses? If indeterminate results were added, this figure must have some bias that this study cannot include. 

Minor:

  • Line 24: positive or negative correlation?
  • Lines 306-310: positive or negative correlation?
  • Lines 379-380: positive or negative correlation?
  • Figure 1b: are they significant?

Author Response

Replies to Reviewer 3

The authors claim in this manuscript that severe COVID-19 was associated with increased indeterminate QuantiFERON® -TB Gold Plus Assay rates. Moreover, they showed that these rates were associated with peripheral blood T-lymphocyte depletion. 

The number of participants is high, and therefore conclusions could be relevant. Nevertheless, there are some issues in the manuscript.

Major issues:

  • The majority of the results presented here are not novel, and several published papers showed the same results and conclusions

We agree with the Reviewer about the presence of some results already published in previous works. Part of these results are needed to describe the characteristics of our cohort of patients. Although we tried to reduce the sections of the manuscript describing not novel results, we believe that it was necessary to confirm that inflammation markers, neutrophil counts, N/L ratio were increased in patients with a more severe disease and a worse outcome in our cohort. Furthermore, these data were used for correlations with IFN-γ levels in the Mitogen-Nil condition. In synthesis, we decided to give a comprehensive picture of our cohort, even if some of the results were not completely novel.  

  • Participants are from the first and the second wave. Do the authors determine the viral strains? This could be important for the results' interpretation. 

Unfortunately, we had the information about the SARS-CoV-2 viral strain only for a small subgroup of patients enrolled in this study, and the number was not sufficient to perform reliable statistical analyses and interpretations. We appreciate the comment of the Reviewer, and will consider this suggestion for further studies.   

  • 20 patients were asymptomatic. As the authors said: "Asymptomatic patients were hospitalized in the infectious disease ward because of concomitant clinical conditions that required inpatient care associated with SARS-CoV-2 transmission prevention, although they did not show any sign or symptom related to SARS-CoV-2 infection". These patients with comorbidities but without symptoms could be removed from the analysis, since they can introduce some bias into the study. 

We thank the Reviewer for this comment. We decided to include also asymptomatic patients in the analysis, after considering that COVID-19 clinical manifestations range from asymptomatic conditions to ARDS and death. We believe that the inclusion of asymptomatic patients can help to generalize our results. Furthermore, all the asymptomatic patients included in the analysis fulfilled the inclusion criteria.

To verify the absence of confounders introduced by the 20 asymptomatic COVID-19 patients, we performed a statistical analysis, after removing the asymptomatic patients. We can confirm all the results showed in the manuscript. Moreover, at the multivariable logistic regression, CD3+ T-cell absolute counts and CD4/CD8 ratio remained the two independent predictors of an indeterminate result of the QFT-Plus assay, with odds ratio and p (odds ratio: 0.998 p=0.002 and odds ratio: 1.639 and p<0.001, respectively) comparable to those reported in the paper (odds ratio: 0.998 p=0.005 and odds ratio: 1.546 and p<0.001, respectively).      

  • Line 211: "No statistical or clinically significant differences were observed for TNF-α, after comparing survivors vs non-survivors, or severe vs non-severe COVID-19 patients (Table 2)". However, in Table 2, the comparison of TNF-a between severe and non-severe was <0.001. Could the authors explain this discrepancy?

We agree with the Reviewer, therefore, the sentence was modified as follows (lines 230-232):

“Although TNF-α did not significantly differ after comparing survivors vs non-survivors, its levels were lower in severe than non-severe COVID-19 patients (p<0,001) (Table 2).”

  • What is the explanation for the percentage of the QuantiFERON® -TB Gold Plus Assay positivity in survivors vs non-survivors? (Table 3). % of positivity in survivors =9.6% and % of positivity in non-survivors= 4.1%. The same tendency was seen between non-severe (10.3%) and severe (6.3%) COVID-19 patients. It could be excepted that the individuals positive for TB could have more severe symptoms. Indeed, TB is a factor found to influence the severity of the disease. How could the authors explain this point?

It is true that there was an increased rate of positive results of the QFT-Plus in survivors and non-severe patients. This could happen because non-survivors and severe COVID-19 patients could have an impaired response to the stimulation with specific TB peptides in TB1 and TB2 tubes. This fact is also supported by the reduced levels of IFN-γ production observed in TB1 and TB2 tubes in non-survivors and severe compared to survivors and non-severe COVID-19 patients, as shown in table 3.

This aspect has been described in the paragraph from line 248 to 252. We also comment that the results of the QFT-Plus assay could be not completely reliable in critically ill COVID-19 patients (lines: 394-397).

It is necessary to specify that a positive QFT-Plus assay identifies a condition of Latent Tuberculosis Infection and not active Tuberculosis. However, for all the patients with a positive QFT-Plus assay active TB was ruled out with Chest CT-scan imaging and/ microbiological assays on sputum or bronchoalveolar lavage samples. For these reasons we did not observe increased severity and worse outcomes in QFT-Plus positive patients.      

  • In Figure 1b: Do the levels of IFN-γ in the Mitogen-Nil condition was shown for all patients or only for those with positive and clear responses? If indeterminate results were added, this figure must have some bias that this study cannot include. 

All the results were reported in figure 1b. We believe that only by representing all the results obtained from the entire cohort (including patients with positive, negative or indeterminate QFT-Plus assay) we can correctly represent the progressive reduction in IFN-γ production of the Mitogen-Nil condition in patients with a more severe disease, as represented by the need of progressively increasing respiratory support (ambient air -> Venturi mask -> mask with reservoir -> non-invasive ventilation -> mechanical ventilation).   

Minor:

  • Line 24: positive or negative correlation?

We used the word “direct” to indicate “positive” correlation. We replaced the word “direct” with “positive”.

  • Lines 306-310: positive or negative correlation?

We specified the type of correlation between IFN-γ levels in the Mitogen-Nil condition and the other parameters (line 321-327):

“The levels of IFN-γ in the Mitogen-Nil condition were positively correlated with total lymphocyte absolute counts, total CD3+, CD3+CD4+, CD3+CD8+, CD3negCD16+CD56+ absolute counts and TNF-α plasmatic levels (Spearman’s test p<0.0001 for all the correlations). An inverse correlation between IFN-γ levels in the Mitogen-Nil condition and CD4/CD8 ratio, neutrophil counts, N/L ratio, CRP, D-dimers, fibrinogen and ferritin was also found (Spearman’s test p<0.0001 for all the correlations) (Figure 2).”

  • Lines 379-380: positive or negative correlation?

We replaced the word correlation with association, which was more appropriate in this case.

  • Figure 1b: are they significant?

Yes, they are significant. We added the significant p value in the figure.

Round 2

Reviewer 3 Report

The authors answered the majority of raised points. But still, they could have introduced more of previous works already published in this field. 

Moreover, it feels that hospitalized patients with asymptomatic COVID-19 must be removed from the study. Since they are hospitalized not for COVID-19 but for other pathologies. 

In Figure 1b, it could be interesting to have histograms without the indeterminate results (as a Figure 1c) besides the actual one. 

Author Response

Replies to reviewer 3 (round 2)

Comments and Suggestions for Authors

The authors answered the majority of raised points. But still, they could have introduced more of previous works already published in this field. 

We thank the reviewer for the suggestion. We added two more references to the introduction section, describing two more studies on the association between COVID-19 severity and indeterminate results of the QuantiFERON-TB Gold test. Therefore, references 16,17,21,22,23,24 and 25 refer to works already published in this field. We modified the introduction as follows (lines 67-74):

“Several studies have shown an increased number of indeterminate QFT assay results in patients hospitalized because of SARS‑CoV‑2 [16,17,21–25]. Patients with an indeterminate response are those with compromised immune responses and inadequate ability to be activated by a mitogen control [21,24]. Furthermore, indeterminate results of the QFT assay could predict mortality in COVID-19 patients [21]. In a recent work on a limited number of subjects, it has been shown that patients with severe COVID-19 had a decreased IFN-γ production after mitogen stimulation compared to asymptomatic COVID-19 patients and healthy donors, in an endemic area for tuberculosis [25]”

Moreover, it feels that hospitalized patients with asymptomatic COVID-19 must be removed from the study. Since they are hospitalized not for COVID-19 but for other pathologies. 

We understand the point raised by the Reviewer. However, in previously published studies asymptomatic COVID-19 patients have been included in the analysis (such as in Gupta, A et al. Indian J. Tuberc. 2021). Although these patients were hospitalized for other reasons, they were moved in our ward after the positivity of the SARS-CoV-2 molecular test on a nasopharyngeal swab, and the QFT-Plus was performed on admission in our ward, therefore after SARS-CoV-2 infection. Having this group of patients in our study can give a more reliable picture of COVID-19 clinical forms, which range from asymptomatic to critical disease and death.

To answer to the Reviewer’s observation, we added a Supplementary material section, which includes tables 1-5 with the results and the statistical analyses after removing the 20 asymptomatic patients. All the previously obtained results were confirmed with this new analysis. A sentence was added in the result section (lines 214-215):

“Statistical analyses after excluding asymptomatic patients are shown in supplementary material.”

In Figure 1b, it could be interesting to have histograms without the indeterminate results (as a Figure 1c) besides the actual one. 

As suggested by the reviewer we added the panel 1c to figure 1, which shows histograms after removing IFN-γ production in the Mitogen-Nil condition of COVID-19 patients with an indeterminate result at the QFT-Plus assay. IFN-γ production was confirmed reduced in patients with a more severe disease, as documented by the need of increased oxygen/ventilation support.

A sentence was also added in the result section (lines 273-275):

“Additionally, similar results were obtained after removing from the analysis patients with an indeterminate result at the QFT-Plus assay (p<0.001).”
